# Relationship between Isokinetic Trunk Muscle Strength and Return to Sports Competition after Conservative Therapy in Fresh Cases of Lumbar Spondylolysis: A Retrospective Observational Study

**DOI:** 10.3390/healthcare11040625

**Published:** 2023-02-20

**Authors:** Yuji Hamada, Yu Okubo, Hiroshi Hattori, Takeshi Nazuka, Yuto Kikuchi, Kiyokazu Akasaka

**Affiliations:** 1Graduate School of Medicine, Saitama Medical University, 981 Kawakado, Moroyama 350-0495, Japan; 2Department of Rehabilitation, Kawagoe Clinic, Saitama Medical University, 7-21 Wakitahontyo, Moroyama 350-0495, Japan; 3School of Physical Therapy, Saitama Medical University, 981 Kawakado, Moroyama 350-0495, Japan

**Keywords:** lumbar spondylolysis, adolescent, athlete, isokinetic trunk muscle strength

## Abstract

This study aimed to clarify the relationship between isokinetic trunk muscle strength and return to sporting activities in fresh cases of lumbar spondylolysis treated with conservative therapy. Patients included a total of ten men (age: 13.5 ± 1.7) who were instructed by their attending physicians to stop exercising and who met the eligibility criteria. Isokinetic trunk muscle strength was measured immediately after exercising for the first time (First) and one month (1M). Flexion and extension and maximum torque/body weight ratio were significantly lower First compared to 1M at all angular velocities (*p* < 0.05). Maximum torque generation time was significantly lower for First at 120°/s and 180°/s than at 1M (*p* < 0.05). Correlations with the number of days to return to sports competition were detected at 60°/s for maximum torque generation time (*p* < 0.05, *r* = 0.65). Following conservative treatment for lumbar spondylolysis, it was considered necessary to focus on trunk flexion and extension muscle strength and contraction speed of trunk flexors at the beginning of the exercise period. It was suggested that trunk extension muscle strength in the extension range might be one of the critical factors for returning to sports.

## 1. Introduction

Lumbar spondylolysis is a general term for fatigue fractures in and around the interarticular processes of the lumbar spine. It has been reported to be associated with increased mechanical loading on the interarticular processes of the lumbar spine due to lumbar extension and rotation movements [1]. Moreover, it is one of the most common sports injuries that occurs during adolescence [2,3]. In new cases of lumbar spondylolysis, conservative treatment was the mainstay of treatment; bony fusion is the highest priority since the failure of bony fusion can lead to spondylolisthesis [4,5]. As such, wearing a corset and discontinuing sports competition for bone fusion is a standard conservative treatment [6,7].

Muscle weakness due to immobilization reaches 6% muscle atrophy in the triceps surae muscle at 2 weeks immobilization [8] and 47% isometric muscle weakness in the vastus lateralis at 3 weeks [9]. Research with American football players has shown that weakness in the internal oblique and erector spinae muscles occurs during the off-season [10]. Based on these facts, there is a concern for the loss of trunk muscle strength due to decreased physical activity during the discontinuation of sports competition for bone fusion in patients with lumbar spondylolysis. The medical staff will take the utmost care not to interfere with bone fusion during the rest period. They will provide exercise therapy to prevent muscle weakness so that the patient can return to sports appropriately when it is time to begin exercise again. However, to our knowledge, no previous studies have been conducted that measure the extent to which trunk muscle strength decreases during the resting period. As such, we assume that the exercise therapy includes stretching and trunk strengthening training focused on lower limb flexibility and trunk stability [11,12,13]. The exercise program is determined by each hospital’s policy and each therapist’s professional judgement. 

Lumbar spondylolysis is associated with increased lumbar lordosis, family history, the presence of spina bifida occulta and lower extremity flexibility [14,15,16,17,18]. Reduced trunk endurance has been reported in 75% of lumbar spondylolysis cases. However, there exist few studies of muscle strength associated with lumbar spondylolysis and they remain scarce [18]. The human osteoligamentous lumbar spine has been reported to buckle under compressive loads of approximately 90 N, much lighter than the weight of the upper torso [19]. Hence, stabilizing and safely moving the lumbar region requires the coordinated work of the trunk muscles, the lumbar multifidus and erector spinae muscles, the abdominal muscle group and the transversus abdominis [20]. Due to the nature of lumbar spondylolysis, which is caused by the stress of repetitive extension and rotation of the lumbar spine, most cases of lumbar spondylolysis in adolescents are received during sports competitions [21,22].Isokinetic muscle strength evaluation is desirable when determining a player’s return to sports due to the required muscle kinematics and muscle strength in each sport. However, though there are some studies on isokinetic muscle strength in trunk flexion and extension in patients with low back pain and lumbar disc herniation [23,24], there are no studies on patients with lumbar spondylolysis. This study aimed to clarify the relationship between the results of isokinetic trunk muscle strength after conservative treatment and the return to sports competition in fresh cases of lumbar spondylolysis. In lumbar spondylolysis cases, alterations in trunk muscle strength due to periods of sports cessation and the relationship between trunk muscle strength and return to sports might be helpful in determining a treatment program.

## 2. Materials and Methods

### 2.1. Study Design and Ethical Considerations

This study is a case-control study following the Checklist for Observational Studies in Epidemiology. All data were collected from the patients’ medical records. This study was registered in the UMIN Clinical Trials Registry (UMIN000049464), approved by the Ethics Committee of the Faculty of Health Sciences, Saitama Medical Center (2022-106) and conducted by the principles of the Declaration of Helsinki.

### 2.2. Participants

The present study included patients diagnosed with lumbar spondylolysis from October 2015 to October 2022 who underwent isokinetic trunk muscle strength testing at the direction of their primary care physician. A total of eighteen patients were excluded: fourteen patients with no limitation of movement due to old fractures, one patient who had no rehabilitation and whose return date to sports competition was unclear and three patients who had only one trunk muscle strength measurement performed. Those who had been judged by their attending physicians to have an old fracture by their attending physicians and had no limitation of movement, those whose return to sports competition was unknown because of intervention solely at the time of muscle strength measurement and those who had only had one trunk muscle strength measurement were excluded. Subjects with applicable fresh fracture cases were included in the study. The attending physician placed all subjects in a corset and exercise was discontinued. The survey parameters included basic information such as age, gender, body weight, sports competition, lumbar spondylolysis separation height and unilateral or bilateral separation. In addition, the number of corset days and return to sports competitions were investigated retrospectively using the medical records. In the present study, the number of days to return to sports competition was considered from the time the patient was permitted to recommence exercising by the attending physician to the time the patient participated in a sports competition again. Participation in sporting events was defined as the day ball players jogged and began practicing with a ball at a team practice and track and field athletes flew and began practicing competitively with their teams.

### 2.3. Isokinetic Trunk Muscle Strength 

An isokinetic testing machine (Biodex System 3, Biodex Corp., Shirley, NY, USA) was used to measure isokinetic trunk flexion/extension muscle strength (Figure 1). The Biodex Dual Position Back Ex/Flex Attachment was connected to the dynamometer. This method of measuring isokinetic trunk muscle strength has proven reliable [23]. All subjects performed approximately three minutes of cycling or simple sit-ups as a warm-up before measurements were taken. The starting position was set according to the method suggested by Grabiner et al. and adjustable pads were placed behind the head, upper trunk and pelvis to secure the participant to the back attachment. The upper trunk, pelvis, and thighs were stabilized with straps [23]. The measured range of motion (ROM) of the spine was 65° to 135° and angular velocities of 60°/s, 120°/s and 180°/s were measured. The subject was instructed to perform three trunk exercises at set ROM and three speeds. Before the measurement was taken, it was fully explained to each patient and practiced at each angular velocity. During the measurement, no one discontinued the process due to pain or other incidents. 

The data used were first-time (First) measurements and the data one month (1M) later. The First data measurement was performed on the day when the patient’s doctor ordered a corset off, permitted them to start exercising and instructed them to measure isokinetic trunk muscle strength. 1M data was conducted on the day the attending physician ordered an isokinetic trunk muscle strength measurement, approximately one month after the first measurement. The survey parameters used were maximum torque/body weight ratio, maximum torque generation time, maximum torque angle, maximum work/body weight ratio and the ratio of flexion/extension (F/E) in flexion and extension, respectively. 

### 2.4. Data Processing

A paired *t*-test or Wilcoxon signed-rank test was performed to compare the isokinetic trunk muscle strength data between the First and 1M measurements. Spearman’s rank correlation coefficient or Pearson’s product moment correlation coefficient was performed for both the number of days to return to sports and initial isokinetic trunk muscle strength. All test results were calculated with effect sizes. Cohen’s d was used for paired *t*-test. Effect sizes r was used for Wilcoxon signed-rank test, Spearman’s rank correlation coefficient or Pearson’s product moment correlation coefficient. In addition, post hoc power analysis was performed on the correlation results. Statistical analysis was performed using IBM SPSS Statistics for Mac, Version 29.0 (Armonk, NY, USA: IBM Corp Released 2020), with a significance level of *p* < 0.05.

## 3. Results

Between October 2015 and October 2022, twenty-eight patients were diagnosed with lumbar spondylolysis. Ten patients were included in our retrospective observational study (Table 1). All patients in the study were ordered to discontinue exercise and to wear corsets by their attending physicians. Each accepted conservative treatment. When bony fusion was confirmed by medical doctors, the patient was allowed to take off his corset and began jogging. Alongside this, dynamic core training and movement exercises specific to sports competitions were allowed and the exercise load was gradually increased.

### 3.1. Difference in Muscle Strength between the First and 1M

#### 3.1.1. Angular Velocity of 60°/s

The results for each of the First and 1M isokinetic trunk muscle strength items are shown in Table 2 and Table 3. Extension maximum torque/body weight ratio and flexion maximum torque/body weight ratio and extension maximum work/body weight ratio and flexion maximum work/body weight ratio were significantly lower at First than at 1M in Table 1 (*p* < 0.05). However, no significant differences were found for the other data (*p* < 0.05).

#### 3.1.2. Angular Velocity of 120°/s, 180°/s

Similar to 60°sec, extension maximum torque/body weight ratio and flexion maximum torque/body weight ratio and extension maximum work/body weight ratio and flexion maximum work/body weight ratio were significantly lower for First than for 1M. In addition to these data, flexion maximum torque generation time was significantly shorter at First than at 1M (*p* < 0.05). 

### 3.2. Correlation between Isokinetic Trunk Muscle Strength at First and Return to Sports Competition Days

#### 3.2.1. Angular Velocity of 60°/s

The correlation between the First isokinetic trunk muscle strength measurement and the number of days to return to sports competition is shown in Table 4, Table 5 and Figure 2. Significant correlations were observed for maximum extension torque generation time (*r* = 0.65) and maximum extension torque exertion angle (*r* = −0.67). No significant correlations were found in the flexion data (*p* > 0.05).

#### 3.2.2. Angular Velocity of 120°/s, 180°/s

As opposed to 60°/s, only the F/E ratio showed a significant correlation (120/s; *r* = 0.71, 180/s; *r* = 0.66). No significant correlations were found for items other than the F/E ratio (*p* > 0.05).

## 4. Discussion

The present study is, to our knowledge, the first to measure isokinetic trunk muscle strength for lumbar spondylolysis. The comparison between First and 1M shows that at an angular velocity of 60°/s, extension maximum torque/body weight ratio and flexion maximum torque/body weight ratio and extension maximum work/body weight ratio and flexion maximum work/body weight ratio were significantly lower at the First than at 1M (*p* < 0.05). However, no significant differences were found in muscle performance, such as angle of exertion and time. The peak value of isokinetic trunk muscle force is inversely related to angular velocity [25]. An angular velocity of 60°/s is the method with the slowest movement speed and the highest peak muscle force value in the present measurements. Belavý et al. reported an 18.3% decrease in transversus abdominis muscle thickness and a 10.6% decrease in internal oblique abdominal muscles after 54 days of rest [26]. Moreover, it has been shown that immobilized rest induces plastic changes in the alterations in the behavioral properties of motor neurons and spinal interneuron circuits [27,28]. Such factors were related, and it was thought that lumbar spondylolysis cases could cause muscle weakness during the exercise cessation period. Iwaki et al. reported that lumbar spondylolysis was associated with decreased muscle endurance of the abdominal muscles in 75% of cases and decreased muscle strength of the abdominal muscles in 70% of cases [18]. Exercise therapy for lumbar spondylolysis often includes trunk and lower extremity stretching, core stability training and strength training of the abdominal musculature [22,29]. Changes with rest have been reported to significantly decrease muscle thickness in the abdominal muscles, a trunk flexor muscle group, but no significant difference in the erector spinae, a trunk extensor muscle group [26,30]. With this considered, the trunk flexor muscle groups may be regularly used in exercise therapy while training for the trunk extensor muscle groups is rarely incorporated. Since both the trunk extensor and flexor muscle groups showed a decrease in muscle strength in First compared to 1M, an exercise program for the trunk extensor muscle groups was considered to be an important inclusion in exercise therapy.

At an angular velocity of 60°/s, 120°/s and 180°/s, there was a significant difference in extension and flexion maximum torque/body weight ratio and extension and flexion maximum work/body weight ratio. In addition, the maximum flexion torque generation time was significantly lower for the First compared to 1M. Flexion maximum torque generation time reflects the speed at which the trunk flexor muscle group reaches maximum contraction. Previous studies have reported that a rest period decreases the contractile rate of Type I and IIa fibers [31]. Since the measurement method used in this study for flexion maximum torque generation time is the activity of the entire trunk flexor muscle group, the results include the muscle activity of Type I and IIa fibers. The results of this study support previous research and suggest that the onset of lumbar spondylolysis and the period of exercise cessation may cause muscle weakness and a decrease in the contraction rate of the abdominal muscle group. Therefore, when isokinetic trunk muscle strength is measured in fresh cases of lumbar spondylolysis with exercise suspension, the maximum torque generation time should be one of the measurements that should be carefully observed.

In this study, the results of the First are inferred to represent some of the muscle strength characteristics of lumbar spondylolysis cases after a period of exercise discontinuation. In previous reports, detailed data were presented for ages 20 years and older and not for adolescents [24,32]. Some studies have been conducted on adolescents. However, the results may differ from the present results because extension and flexion maximum torque were not calculated with body weight ratio in the data of healthy subjects [33]. Comparison of isokinetic trunk muscle strength in adolescent lumbar spondylolysis cases from previous studies is difficult, suggesting that the data presented in this study may serve as a reference for isokinetic trunk muscle strength in lumbar spondylolysis cases after the exercise cessation period is over.

The second outcome was an investigation of the correlation between the First isokinetic trunk muscle strength and the number of days to return to sports competition. Angular velocity of 60°/s showed significant correlations with maximum extension torque generation time (*r* = 0.65) and maximum extension torque exertion angle (*r* = −0.67). Extension muscle strength is measured from the most trunk-flexed position of the angle set in relation. Therefore, the maximum extension torque exertion angle and the maximum extension torque generation time are inferred to be related in function. This suggests that patients who take longer to return to work may be characterized by a muscle characteristic that exerts maximum muscle strength in the extension range during trunk extension movement. The direction of stress on the vertebral arch, common in lumbar spondylolysis, has been reported during trunk rotation and extension [1]. As such, it was suggested that the ability to exert maximum trunk extension muscle strength in the trunk flexed position may be one of the abilities required for an early return to sports competition. 

Furthermore, for angular velocities of 120/s and 180/s, a significant correlation was established between the number of days from the start of exercise to the return to the sports competition and the F/E ratio (120/s; *r* = 0.71, 180/s; *r* = 0.66). The F/E ratio is lower with higher trunk extensor strength. Therefore, at angular velocities of 120°/s and 180°/s, trunk extensor muscle strength values were high, suggesting that the ratio of flexion and extension is a factor associated with return to sports competition. Data at an angular velocity of 60°/s also revealed an association between return to sports competition and trunk extensor muscle group. Lumbar spondylolysis patients were reported to show lower fast-twitch motor unit recruitment in the erector spinae compared to controls [34]. Hence, it is possible that return to sports should focus on trunk extensor muscle strength than trunk flexor muscle strength. Moreover, muscle imbalances have been reported to be associated with injury and the ratio of extension and flexion muscle strength is important [35,36]. This study design cannot detect a target F/E ratio. However, it suggests that the ratio of trunk extension and flexion groups may also be important in return to sports competition days in lumbar spondylolysis cases. 

This study had several limitations. The findings of this study are from a retrospective observational study. The data comparing the First to 1M does not clarify whether the muscle weakness that existed before the injury was a result of the improvement from exercise therapy or whether the muscle weakness due to rest is a result of the improvement from the resumption of exercise. To clarify these findings, comparisons with subjects who did not discontinue exercise and underwent conservative treatment and healthy individuals of the similar age group are needed. Further developmental studies should be conducted in the future. Second, pain and other physical functions during the period of a full return to sports, performance and recurrence have yet to be investigated. Investigating and clarifying the relationship between these and other data will further enhance the significance of measuring isokinetic trunk muscle strength. Finally, the number of subjects was limited. Therefore, a power analysis of the posterior test was conducted. The results were (1–β = 0.58) for the maximum extension torque generation time at 60°/s, which has the lowest correlation, and (1–β = 0.70) for the F/E ratio at 120°/s, which has the highest correlation. The general recommended β error is 0.8 and it is necessary to consider that the value of β error is small.

## 5. Conclusions

This research investigated the history of trunk extension and flexion muscle strength in fresh cases of lumbar spondylolysis. After the exercise cessation period, trunk extension and flexion muscle strength at an angular velocity of 60°/s were significantly lower the first time of measurement than at 1 month. Angular velocities of 120°/s and 180°/s showed a similar trend to that of 60°/s and the maximum flexion generation time was also significantly lower. The results suggest that it is necessary to include an exercise program to prevent muscle weakness in the trunk flexor and extensor muscle groups even during periods of rest from sports competition. In addition, a correlation was recognized between the number of days of return to the sports competition and the angle of maximum trunk extension torque exertion at an angular velocity of 60°/s maximum torque generation time and maximum torque angle. This suggests the need to focus on trunk extension strength as one of the factors related to the return to sports competition.

## Figures and Tables

**Figure 1 healthcare-11-00625-f001:**
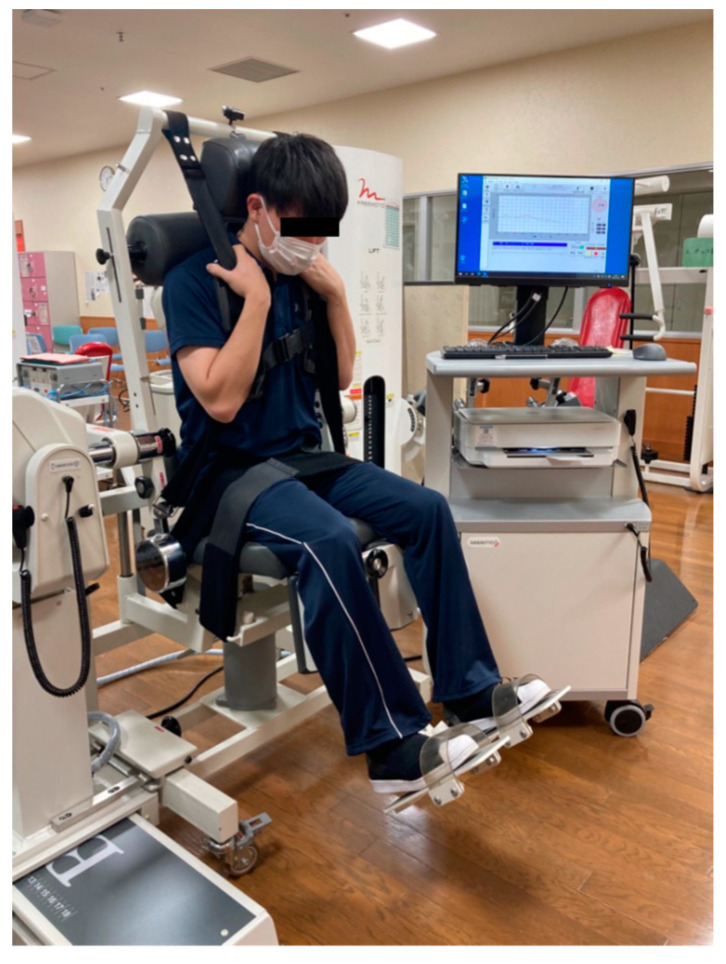
Measurement isokinetic trunk flexion/extension muscle strength.

**Figure 2 healthcare-11-00625-f002:**
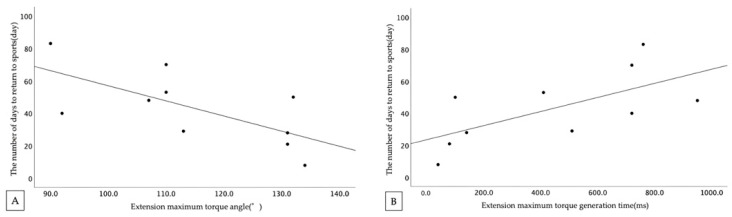
Correlation between days to return to sports competition and angular velocity 60°/s. (**A**) maximum extension torque generation time, (**B**) maximum extension torque exertion angle.

**Table 1 healthcare-11-00625-t001:** Basic attributes of characteristics the participants (*n* = 10).

Variable	
Sex, *n*; Male/Female	10/0
Age (years)	13.5 ± 1.7
body weight (kg)	56.5 ± 9.1
The level of separation	L4; 2, L5; 8
Separation Characteristics, unilateral/bilateral	5/5
Sports competition, *n*; baseball/soccer/track and field/volleyball/basketball	3/4/1/1/1
Corset wearing period (days)	56.0 ± 17.0
Corset off to return to sports competition (days)	42.8 ± 22.7

**Table 2 healthcare-11-00625-t002:** Comparison of isokinetic extension trunk muscle strength between the First and 1M.

	First (*n* = 10)	1M (*n* = 10)	*p*-Value	95% CI		Effect Size
60°/s						
maximum torque/body weight ratio	88.1 ± 27.59	125.9 ± 33.61	0.003 *	−59.25	−16.37	1.26
maximum torque generation time	443.0 ± 336.68	293.2 ± 198.96	0.223	−115.30	414.90	0.40
maximum torque angle	115.0 ± 16.45	116.2 ± 20.20	0.877	−18.31	15.91	0.05
maximum work/body weight ratio	78.7 ± 25.79	104.6 ± 32.64	0.004	−41.10	−10.56	1.21
120°/s						
maximum torque/body weight ratio	71.4 ± 16.28	116.5 ± 34.66	<0.001 *	−64.13	−26.05	1.69
maximum torque generation time	235.0 ± 119.19	194.0 ± 101.67	0.318	−46.64	128.64	0.34
maximum torque angle	115.0 ± 18.38	117.7 ± 12.84	0.615	−14.44	9.04	0.17
maximum work/body weight ratio	63.7 ±18.34	92.5 ± 31.86	0.003 *	−44.70	−12.74	1.29
180°/s						
maximum torque/body weight ratio	59.0 ± 17.07	95.57 ± 41.91	0.010 *	−61.96	−11.26	1.03
maximum torque generation time	213.0 ± 82.7	168.0 ± 72.23	0.080	−6.61	96.61	0.62
maximum torque angle	107.8 ± 13.37	112.3 ± 13.28	0.269	−13.15	4.15	0.60
maximum work/body weight ratio	40.1 (34.9−43.5)	84.7 (64.6−139.4)	0.037 *	-	-	0.66

Statistical analysis was performed using a paired *t*-test or Wilcoxon signed-rank test. * The values indicate statistical significance of *p* < 0.05. For normally distributed data, mean ± standard deviation (SD); for non-normally distributed data, median (25−75 tile). The effect size is Cohen’s d for paired *t*-test and r for Wilcoxon signed-rank test.

**Table 3 healthcare-11-00625-t003:** Comparison of isokinetic flexion trunk muscle strength between the First and 1M.

	First (*n* = 10)	1M (*n* = 10)	*p*-Value	95% CI		Effect Size
60°/s						
Flexion/Extension ratio	67.4 (53.4−81.5)	65.5 (57.6−77.2)	0.959	-	-	0.16
maximum torque/body weight ratio	59.6 ± 11.44	84.1 ± 18.14	0.002 *	−36.85	−12.17	1.42
maximum torque generation time	435.0 (130.0−680.0)	170.0 (150.0−320.0)	0.168	-	-	0.44
maximum torque angle	82.5 (73.0−100.0)	73.5 (72.0−76.0)	0.168	-	-	0.44
maximum work/body weight ratio	50.8 ± 13.28	60.8 ± 10.73	0.002 *	−15.34	−4.62	1.33
120°/s						
Flexion/Extension ratio	75.7 (49.8−78.3)	61.4 (56.5−63.6)	0.333	-	-	0.31
maximum torque/body weight ratio	46.7 (39.8−57.0)	64.1 (57.4−85.7)	0.005 *	-	-	0.89
maximum torque generation time	345.0 (280.0 - 450.0)	180.0 (170.0−250.0)	0.028 *	-	-	0.69
maximum torque angle	95.2 ± 14.80	86.2 ± 10.08	0.201	−5.76	23.76	0.44
maximum work/body weight ratio	38.4 ±15.10	54.8 ± 15.47	<0.001 *	−22.63	−10.15	1.88
180°/s						
Flexion/Extension ratio	70.2 (64.6−77.1)	55.5 (48.4−85.10)	0.575	-	-	0.18
maximum torque/body weight ratio	41.0 ± 13.87	55.5 ± 14.01	0.005 *	−23.50	−5.46	1.15
maximum torque generation time	352.0 ± 124.79	268.0 ± 98.52	0.030 *	10.03	157.97	0.81
maximum torque angle	112.0 (98.0−114.0)	102.0 (97.0−111.0)	0.415	-	-	0.26
maximum work/body weight ratio	27.6 ± 13.99	39.8 ± 16.29	<0.001 *	−16.85	−6.35	1.58

Statistical analysis was performed using a paired *t*-test or Wilcoxon signed-rank test. * The values indicate statistical significance of *p* < 0.05. For normally distributed data, mean ± SD; for non-normally distributed data, median (25−75 tile). The effect size is Cohen’s d for paired *t*-test and r for Wilcoxon signed-rank test.

**Table 4 healthcare-11-00625-t004:** Correlation between First isokinetic extension trunk muscle strength measurement and the number of days to return to sports competition.

	*p*-Value	*9*5%CI		*r*-Value	Power (1-β)
60°/s					
maximum torque / body weight ratio	0.674	−0.53	0.71	0.153	0.07
maximum torque generation time	0.041 *	0.04	0.91	0.651 **	0.62
maximum torque angle	0.033 *	−0.92	−0.08	−0.674 **	0.58
maximum work / body weight ratio	0.348	−0.38	0.80	0.332	0.154
120°/s					
maximum torque / body weight ratio	0.667	−0.72	0.52	−0.156	0.07
maximum torque generation time	0.904	−0.60	0.66	0.044	0.051
maximum torque angle	0.683	−0.53	0.71	0.148	0.068
maximum work / body weight ratio	0.621	−0.73	0.51	−0.179	0.077
180°/s					
maximum torque / body weight ratio	0.828	−0.58	0.68	0.079	0.055
maximum torque generation time	0.431	−0.77	0.42	−0.281	0.121
maximum torque angle	0.286	−0.33	0.81	0.375	0.188
maximum work / body weight ratio	0.869	-	-	0.06	0.052

Analysis carried out using Pearson Correlation Coefficient or Spearman’s rank correlation. * The values indicate statistical significance of *p* < 0.05. ** Correlation coefficients for items with significant differences in *p*-Values.

**Table 5 healthcare-11-00625-t005:** Correlation between First isokinetic flexion trunk muscle strength measurement and the number of days to return to sports competition.

	*p*-Value	*9*5%CI		*r*-Value	Power (1-β)
60°/s					
Flexion/Extension ratio	0.952	−0.62	0.64	0.022	0.05
maximum torque/body weight ratio	0.281	−0.33	0.81	0.379	0.191
maximum torque generation time	0.967	−0.64	0.62	−0.015	0.05
maximum torque angle	0.527	−0.47	0.75	0.227	0.095
maximum work/body weight ratio	0.159	−0.21	0.85	0.481	0.301
120°/s					
Flexion/Extension ratio	0.021 *	0.15	0.93	0.712 **	0.703
maximum torque/body weight ratio	0.076	−0.07	0.89	0.584	0.457
maximum torque generation time	0.843	−0.67	0.58	−0.072	0.054
maximum torque angle	0.131	−0.17	0.86	0.511	0.342
maximum work/body weight ratio	0.073	−0.06	0.89	0.59	0.468
180°/s					
Flexion/Extension ratio	0.038 *	0.05	0.91	0.659 **	0.597
maximum torque/body weight ratio	0.154	−0.20	0.85	0.486	0.308
maximum torque generation time	0.424	−0.78	0.42	−0.285	0.124
maximum torque angle	0.333	−0.37	0.80	0.342	0.161
maximum work/body weight ratio	0.181	−0.24	0.84	0.46	0.27

Analysis was carried out using Pearson Correlation Coefficient or Spearman’s rank correlation. * The values indicate statistical significance of *p* < 0.05. ** Correlation coefficients for items with significant differences in *p*-Values.

## Data Availability

The data presented in this study are available on request from the corresponding author.

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
