# Peer review of "Relationship between Isokinetic Trunk Muscle Strength and Return to Sports Competition after Conservative Therapy in Fresh Cases of Lumbar Spondylolysis: A Retrospective Observational Study"

_healthcare, 2023, doi:10.3390/healthcare11040625_

Round 1

Reviewer 1 Report

The English should be carefully checked.

In the full paper: please check carefully the punctuation because, as it is,  the reader cannot clearly realize several sentences.

Please mention every Figure and Table in the text before they appear.

The paper does not provide any literature review,

Research goals are to be presented. 

Propose future work with research gaps.

Author Response

Dear Reviewer,

Thank you for your comments and suggestions. We have revised the manuscript carefully, based on your comments.

I would be grateful if you could confirm the attached file for the revised content.

Sincerely yours,

Saitama Medical University Graduate School of Medicine
Yuji Hamada

Reviewer 2 Report

There is a need to improve the study methodology:

Inclusion criteria: No data are provided on when the fracture occurred and the day that the test was performed after a suspected fracture.

fresh fracture must be defined in terms of number of days to be diagnosed. Arthrogenic inhibition begins shortly after trauma. It is not explained in the methodology how many days after the trauma or the symptoms the corset was used.

This exclusion criterion should be better explained: Improve this phasis “those whose return to sports competition was unknown because of intervention only at the time of muscle strength measurement”

What determined the suspension of the use of the corset?

I believe that the correlation between trunk extension muscle strength and the number of days needed to return to competitions is difficult to achieve, as athletes perform different modalities and the preparation for returning to sport is different for each sport modality. The authors did not describe the type of rehabilitation performed, which may be an intervening factor in the study result.

I assess that in the current format it is not possible to accept the paper.

It would be necessary to modify the background, objectives,  methodology and consequently  results and discussion.

Author Response

(The authors gave the same response as above.)

Reviewer 3 Report

It would be interesting to know the average consolidation time of this specific type of participant - does it differ at all from the other adolescents?

please make it clear at the outset that the study was based on the observation of medical records. It may only be necessary to change the order of the sentences.

In line 94 they only talk about "ball players", were there any other type of participants practising another sport modality? Can the sport modality influence their results? Could athletes playing more contact sports, such as rugby, be included in this study?

In line 103, add the bibliography of Grabiner et al.

Change the sentence: "Eighteen patients who met the exclusion criteria were excluded" from line 131 to the methodology. It would be interesting if you could make a flow chart of participant selection.

Figure 2 does not look good.

Please be clearer in the conclusions.

Author Response

(The authors gave the same response as above.)

Round 2

Reviewer 1 Report

The authors have incorporated the provided suggestions

Reviewer 2 Report

After the modifications made by the authors, I believe that the article can be accepted for publication in this journal.